# Frequency of relapse for severe acute malnutrition and associated factors among under five children admitted to health facilities in Hadiya Zone, South Ethiopia

**Abera Lambebo**[1]*, **Deselegn Temiru**[2], **Tefera Belachew**[2]

**1** Department of Public Health, Institute of Medical and Health Science, Collage of Health Science, Debre Berhan University, Debre Berhan, Ethiopia, **2** Department of Nutrition and Dietetics, Faculty of Public Health, Jimma University, Jimma, Ethiopia

* lambebo70@gmail.com

**Data Availability Statement:** All relevant data are within the manuscript and its Supporting Information files.

## Abstract

### Background

Severe acute malnutrition is a common cause of morbidity and mortality among under five children in Ethiopia. A child may experience more than one episode of SAM depending on the improvement of the underlying factors. However, there is no study that determined the frequency of relapse of SAM cases after discharge in Ethiopia.

### Objective

To identify the frequency of relapse and associated factors among children discharged after undergoing treatment for SAM in Hadiya Zone, South, Ethiopia.

### Methods

An institution based retrospective cohort study was done among children admitted to health posts for treatment of SAM from 2014/2015-2019/2020 under-five children's after discharge in health post for severe acute malnutrition in the last five years in Hadiya zone, SNNPR, Ethiopia. Both first admission data and relapse data were abstracted from the records of the SAM children from Aguste 1–30 /2020 Using a data collection format. Data were coded and edited manually, then doubly entered into Epi-Data statistical software version 3.1 and then exported to SPSS for windows version 26. After checking all the assumptions finally Negative binomial regression for poison has been used. All tests were two sided and P values <0.05 were used to declare statistical significance.

### Results

In the last five year there were the proportion of relapsed cases were 9.6%, 95% CI: (7.7%, 11.7%) On multivariable negative binomial regression model, after adjusting for background variables relapse of severe acute undernutrition was significantly associated with having edema during admission with (IRR = 2.21, 95% CI:1.303–3.732), being in the age group of 6–11 months (IRR = 4.74,95% CI:1.79–12.53), discharge MUAC for the first admission (P =

**Funding:** For this study material and finical support is got from Jimma University post graduate school as PhD student research support fund. The role the funders took in the study, Funders had no role in this study as I am PhD candidate in the above-mentioned university funders has no role except giving support for their students on the time of research. And no authors mentioned in this study had received any salary for this particular study as the last two authors were from Jimma University, and they are employee of Jimma university. In this study including myself all authors received no specific funding for this work.

**Competing interests:** The authors have declared that no competing interests exist.

0.001, IRR = 0.37, 95% CI:0.270–0.50) increase the risk of incidence rate ratio(IRR) relapse case of severe acute under nutrition.

## Conclusion

Frequency of SAM relapse was positively associated with age, having edema during admission, while it was negatively associated with discharge MUAC. The results imply the need for reviewing the discharge criteria taking into account the recovery of MUAC as a marker for lean tissue accretion, especially in edematous children and those in the younger age.

## Introduction

### Background

Malnutrition referring to deficiencies, excesses, or imbalances in a person's intake of energy and/or nutrients and it affects every country in the world by one or more forms [1]. Severe acute malnutrition(Wasting) is characterized by low weight-for length/ height and or bilateral pitting edema [2]. Severe acute malnutrition [3] has several immediate, underlying and basic causes. Once a child develops SAM, he/she often suffers from chronic and lifelong consequences throughout life continuing the miserable legacy from generation to generation [4].

Nearly half of all deaths in children under 5 are attributable to undernutrition which puts them at greater risk of dying from common infections and delays recovery from them [5]. Large proportion of childhood malnutrition occur mainly among under five children living in low-income and middle-income countries [6]. In Africa there are large burden of risk factors related to childhood health and development, most of which are of an infective or social origin [7].

It is manifests by either marasmus, Kwashiorkor, marasmic kwashiorkor or non-edematous malnutrition is form of severe under nutrition, the child is severely wasted and has the appearance of "skin and bones" due to loss of muscle and fatty tissue. The child's face looks like an old man's following forfeiture of facial subcutaneous fat, but the eyes may be watchful and the ribs are visible. There might be folds of skin on the buttocks and thighs that make it look as if the child is wearing "baggy pants Children with severe acute under nutrition has very low weight for their height and severe muscle wasting and they may also have nutritional edema–characterized by swollen feet, face and limbs [8, 9].

Although relapse of SAM is one of the problems encountered in the management of children with severe acute malnutrition, its magnitude and associated factors are not documented so far.

Study in rural Malawi among children 6 to 59 months old with MAM shows that mid-upper arm circumference and WHZ at the end of supplementary feeding were the most important factors in predicting which children remained well-nourished [10].

In Ethiopia, children with SAM are admitted to health posts using MUAC < 11.5cm and get treated with ready to use therapeutic food and other treatments indicated in the protocol for a period of 8 weeks (Refer to the national SAM Guideline). Children with the lowest MUAC at admission showed a significant gain in MUAC but not weight, and children with the lowest weight-for-height/length (WHZ) showed a significant gain in weight but not MUAC and response to treatment was largest for children with the lowest anthropometric status at admission in either measurement modality by WHZ or MUAC [11].

However, they are discharged from the program based on percent of weight gained or weight for height > 70%. During discharge recovery based on MUAC is not used. We

suspected whether discharge before full recovery MUAC could lead to increased frequency of relapse of SAM.

There is one case control study conducted in Ethiopia to identify the determinant factors for relapse [12] however, this study was limited to factor identification and fail to identify frequency of relapse for this mater this study considers case count of relapse cases in selected area and time. So, this study is aimed to identify the number of relapse cases and associated factors.

## Methods

An institution retrospective cohort study was conducted among the cohort of 2014/2015-2019/2020 under-five children's who are admitted and discharged for SAM case in 20 selected health posts in Hadiya zone, SNNPR, Ethiopia from Aguste 1–30 /2020.

According to a May 24, 2004 World Bank memorandum, 6% of the inhabitants of Hadiya have access to electricity, this zone has a road density of 104.1 kilometers per 1000 square kilometers compared to the national average of 30 kilometers) [13], the average rural household has 0.6 hectare of land compared to the national average of 1.01 hectare [14] the equivalent of 0.6 heads of livestock. 22.8% of the population is in non-farm related jobs, compared to the national average of 25% and a Regional average of 32%. 74% of all eligible children are enrolled in primary school, and 21% in secondary schools. 43% of the zone is exposed to malaria and the memorandum gave this zone a drought risk [15]. This zone is characterized by a predominant commitment to agricultural activities, especially the *enset*-growing, which is often combined with that of grain, barley and maize, as well as the breeding of domestic animals [16].

In Hadiya Zone there were 280 Health Posts (HPs), 60 rural Health Centers, one University teaching Hospital and 3 primary level Hospitals. Hadiya zone is divided into 11 districts for administrative purposes. The vast majority of the population are Hadiya in ethnic group and they earn their living through rain fed agriculture and it has 12 woradas and 2 administrates towns. The woradas were; East Bedewacho, Siraro Bedewacho, West Bedewacho and Shone town administration separated from the rest of the zone by Kembeta Tambaro and the administrative center of Hadiya is Hosanna [17]. Of which this was study conducted in two woradas and one town administration among 20 health posts with highest number of cases East Bedawacho (Tikere kokere,Tikare Anbesa,Mahal,Jariso,Amburse Anjulo,2nd Chafa,Eddo,Lenda, Jerso Kutube and Bente Wosen).

Siraro Bedewacho (Abuka,Langano,Dongaro Bonkoya,Wera Bonkoya,sheriko Gafarso, Kumudo,Beshilo,Mahal Korga and Woldia) and Shone town administration (Wera Gere and Shone City Adimin). And health posts were selected based on number of SAM cases.

### Study population

Documents of all under five children that were admitted for severe acute undernutrition in selected health posts in the last five year.

### Inclusion

All documents that full registered about the admission and discharge status were included for this study.

### Exclusion criteria

Documents that double registered or registration after transfer for other facilities were excluded.

## Sample size calculation

Sample size for this study is all cases with in selected area at fixed time that means number of relapse cases among admitted children for severe acute malnutrition in health posts of two woradas and one town administration in the last five years.

## Sampling technique

For this study all severe acute malnutrition cases those admitted in selected woradas and health posts were included and woradas and health posts were selected conveniently based on their case load.

## Data collection methods

For collecting data from the registration book of under-five children with SAM, structured list of questioners was used during the survey for relapsed cases of SAM in the last five years. And the questionaries were adopted from previous study that was conducted in Malawi for similar topics [7]. To ensure data quality, a three days training was given for data collectors and supervisors on the data collection tool, the data collection procedure and questionnaire was pretested on children with SAM in Halaba, which is not part of the study area.

## Operational definition

Relapse rate/repeated relapse episodes; The proportion of children who re-enrolled after they recovery and discharged [18].

Wasting is defined; as low weight-for-height. It often indicates recent and severe weight loss, although it can also persist for a long time [1].

Severe acute malnutrition; It is diagnosed by weight for- height below -3 SD of the WHO standards, by a MUAC < 11.5 cm and by Clinical sign having bilateral edema [8, 19, 20].

Kwashiorkor or edematous malnutrition; is also form of severe under nutrition, the child's muscles were wasted, but wasting may not be apparent due to generalized edema or swelling from excess fluid in the tissues [8, 9].

Criteria for discharging children from treatment; weight-for-height/length is $\geq -2$ Z-scores and they have had no oedema for at least 2 weeks [21].

## Data processing and analysis

The data were doubly entered by two data clerks into Epi-Data version 3.1 to avoid clerical errors using side by side comparison. and the data were then exported to SPSS for windows version 26 statistical software for cleaning and analysis. Descriptive analysis such as simple frequencies, measures of central tendency, and measures of variability was used to describe age and sex distribution as well as discharge status of the under-five children for severe acute malnutrition treatment.

Before poison regression the assumptions were checked, as the variable is relapse case count it meets the first assumption for poison regression. Then One-Sample Kolmogorov-Smirnov Test was done to check significance test for non-significant value and as result reveals 0.978 so this data full fills the second assumptions again when we see the distribution of the data for third assumptions; mean = 0.1178 and variance = 0.149 these values indicate the over-dispersion of data, for as we looking for poison regression another assumption of poison regression is equi-dispersion of data. However, for this data it fails to meet the last assumption of poison regression. As the last assumption fails, we have conducted **negative binomial regression** for poison. A negative binomial regression selection of variables for multiple

negative binomial regression is based on P-value <0.25 and final significance for Incidence Risk Ratio (IRR) was declined at a P-value of < 0.05.

### Ethical considerations

Before starting the data collection process, ethical clearance was be secured by Jimma University Health Research Ethics Review Committee (IHRERC). An official letter was written from Jimma University to the Hadiya Zone health office.

Informed written consent was obtained from all health extension workers of selected health posts and woreda health office, confidentiality of the study documents was' information was also ensured according to the Helsinki declaration of ethical code for human subjects.

### Results

In this study, the relapse case count has been conducted for severe acute malnutrition in two woredas and one town administrative in 20 health posts among 900 children with severe acute malnutrition in the last five years before the survey. From the total case counts from the records 465(51.7%) were females and 435(48.3%) were males. The mean (±sd) age of the children in this study was 26.1±0.496 months. Regarding the types of admissions, from 900 children with SAM, 814(90.1%) were new admissions (Table 1).

From the total admissions for SAM, 575 (63.9%) were non-edematous diagnosed as marasmic cases, while and 325(36.1%) were edematous diagnosed as kwashiorkor and the rest were diagnosed as marasmic kwashiorkor. The mean (± SD) days of stay on treatment after admission was 46 days (±12.98) days. Regarding the treatment outcome of admitted children, 838 (93.1%) were cured, 6 (0.7%) died while 20(2.2%), 16(1.8%),14(1.6%), and 6(0.7%) were Defaulter, Unknown, Non-response, and medical transfer cases, respectively (Table 2).

Of the total case treated in the20 health posts, 86 (9.6%), 95% CI: (7.7%, 11.7%) of SAM cases were readmitted with similar cases in the last five year out of which 66 children were readmitted once and the rest 20 cases were readmitted twice (Fig 1).

**Table 1. Socio demographic characteristic of under five children who are admitted for severe acute under nutrition in Hadiya Zone, SNNPR, Ethiopia in the las five years from 2014/2015-2019/2020 (n = 900).**

| Variable | Frequency | Percent |
|---|---|---|
| Sex | | |
| Male | 435 | 48.3 |
| Female | 465 | 51.7 |
| Age, Months | | |
| 6–11 | 206 | 22.9 |
| 12–23 | 171 | 19 |
| 24–35 | 161 | 17.9 |
| 36–47 | 226 | 25.1 |
| 48–60 | 136 | 15.1 |
| Residential worada | | |
| Siraro Bedewacho | 507 | 56.3 |
| East Bedewacho | 307 | 34.1 |
| Shone City administration | 86 | 9.6 |
| Types of admission | | |
| New admission | 814 | 90.4 |
| Re admission | 86 | 9.6 |

**Table 2. Nutritional status of the children during admission among severe acute under nourished under five children in Hadiya Zone, SNNPR, Ethiopia in the las five years from 2014/2015-2019/2020 (n = 900).**

| Nutritional status and diagnosis | Number | Percent |
|---|---|---|
| **Presence of edema** | | |
| Yes | 325 | 36.1 |
| No | 575 | 63.9 |
| **Diagnosis during admission** | | |
| Marasmus | 575 | 63.9 |
| Kwashiorkor | 319 | 35.4 |
| Marasmic kwashiorkor | 6 | - |
| **Treatment out comes** | | |
| Cured | 838 | 93.1 |
| Dead | 6 | - |
| Defaulter | 20 | 2.2 |
| Unknown | 14 | 1.6 |
| Transfer out | 16 | 1.8 |
| Non response | 6 | - |
| **MUAC of the children when termination treatment(cm)** | | |
| <11.5 | 270 | 30 |
| 11.5 <-12.5 | 396 | 44 |
| >12.5 | 175 | 19.4 |
| Not recorded | 59 | 6.6 |

From the total of 86(9.6%,) 95% CI: (7.7%, 11.7%) relapsed cases 44 (4.9%) were males and the rest 42(4.7%) were females. The outcome for the first admission showed that 48(5.3%) were cured and discharged for the first admission 20(2.2%) were defaulters for the first admission 10(1.1%) was with unknown status and the rest 8 were transferred out for a medical reason. Regarding the age of relapsed cases 34(39.5%) were at the age of 6–11 months followed by those in the age group of 36-47(23.3%) and 12–23 months (15.1%) (Table 3).

On multivariable Negative binomial regression, after adjusting for background variables including sex of the child, admission edema, child age, discharge weight for age ratio Z score [8], discharge mid-upper arm circumference (MAUC) and a number of days in treatment having edema at admission, age of the child, MUAC at discharge, and having edema on the first admission were independent predictors of relapse.

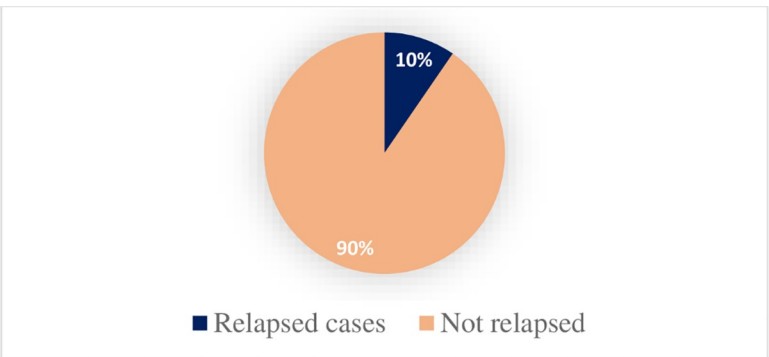

**Fig 1. The number of relapsed cases of severe acute under nourished under five children in Hadiya Zone, SNNPR, Ethiopia from 2014/2015-2019/2020 (n = 900).**

**Table 3. Relapsed cases with other variables severe acute under nourished under five children in Hadiya Zone, SNNPR, Ethiopia from 2014/2015-2019/2020 (n = 900).**

| Variable | Relapse of SAM cases among under five children | |
|---|---|---|
| | Yes | No |
| | n (%) | n (% |
| Sex | | |
| Male | 44 (4.9) | 391(43.4) |
| Female | 42(4.7) | 423(47) |
| Admission edema | | |
| Yes | 32(3.6) | 293(32.6) |
| No | 54(6) | 521(57.9) |
| Outcome for the first admission | | |
| Cured | 48(5.3) | 790(87.8) |
| Defaulter | 20(2.2) | 0 |
| Non responses | 0 | 6 |
| Transfer out | 8 | 8 |
| Unknown status | 10(1.1) | 4 |
| Age, months | | |
| 6–11 | 34(3.8) | 172(25) |
| 12–23 | 13(1.4) | 158(17.6) |
| 24–35 | 14(1.6) | 147(16.3) |
| 36–47 | 20(2.2) | 206(22.9) |
| 48–60 | 5 | 131(14.6) |
| Discharge MUAC (cm) | | |
| <11.5 | 53(5.9) | 217(24.1) |
| 11.5–12.5 | 23(2.6) | 373(41.4) |
| >12.5 | 6 | 169(18.8) |
| Not recorded | 4 | 55(6.1) |

Having nutritional edema during the first admission increased incidence rate ratio of relapse for SAM by 2.205 times (IRR = 2.21, 95% CI:1.303–3.732). Similarly, being in the age groups of 6–11 months increased the incidence rate ratio of relapse for SAM by 4.7 times compared to the age group of 48–60 months (IRR = 4.74,95% CI:1.79–12.53). Likewise, having edema during the first admission increased the incidence rate ratio of relapse by 2.2 times ((P = 0.003, IRR = 2.2, 95% CI:1.30–3.73).

Conversely, there was a negative relation between discharge MUAC for the first admission and relapse of SAM. For 1cm increase in the discharge MUAC of the first admission, the incidence rate ration of relapse of SAM decreased by 63% (P = 0.001, IRR = 0.37, 95% CI:0.270–0.50) (Table 4).

## Discussion

We found out that the proportion of relapse was 9.6%, 95% CI: (7.7%, 11.7%) which is in line with study conducted in rural Malawi reveals that children treated for SAM and discharged in 8 weeks 7% relapse after treatment [10]. Similarly, Another Prospective cohort studies in Bangladesh, among severely malnourished children's reveal that from those who treated for severe malnutrition and discharged by weight for height but not for MUAC; 7% required re admission to the nutrition program because of [22]. From this we may conclude that the problem of relapse among severe acute malnourished children is common. However, there is no tracer

**Table 4. Negative binomial regression model for factors associated with incidence rate ratio (IRR) of relapse of SAM case in Hadiya Zone, SNNPR, Ethiopia.** 2014/2015-2019/2020 (n = 900).

| Variable | β | P | IRR (95% CI) |
|---|---|---|---|
| sex | | | |
| Male | 148 | 0.52 | 1.16 (0.74–1.82) |
| Female | | . | 1.00 |
| Edema during admission | | | |
| Yes | 0.79 | 0.003* | 2.20(1.30–3.73) |
| No | | | 1.00 |
| Age, months | | | |
| 6–11 | 1.56 | 0.002* | 4.74(1.79–12.53) |
| 12–23 | 0.93 | 0.093 | 2.53(0.86–7.50) |
| 24–35 | 0.71 | 0.160 | 2.04 (0.76–5.51) |
| 36–47 | 0.68 | 0.148 | 1.984 (0.78–5.02) |
| | | - | 1.00 |
| Discharge MUAC for the first admission | -1.001 | 0.001* | .368 (.27-.50) |
| Number of days in treatment for the first admission | 0.006 | 0.48 | 1.006 (.990–1.02) |

IRR: Incidence rate ratio.

P<0.01.

CI: Confidence interval.

system after discharge from the programs in health system and this may lead to repeated admission of children for similar problem.

On other hand, it was observed that MUAC at discharge of the first admission was associated with incidence rate ratio of relapse. For 1cm increase in the MUAC at the discharge of the first admission from the existing one or 11.5cm it will decrease the incidence rate ratio of relapse by 63%. This finding is in line with the study done in rural Jharkhand and Odisha, eastern India, and in Burkina Faso which showed that as anthropometric indicators were hazardous for MAM and for SAM as MUAC at 11.5 cm [9, 10]. As MUAC is a measure fat free mass, which is mostly lean tissue, it is an indicator of recovery in terms of fat free masa accretion [23]. The findings imply that early discharge of children with SAM before return of MUAC will result in a relapse of severe acute malnutrition as there is still a need for more time before discharge for restoration of the wasted lean tissue. This implies the need for revising the existing cutoff point of MUAC for discharge as anthropometric cure may not guarantee the risk of relapse in SAM cases.

Similarly, having edema during admission increased the incidence rate ratio of relapse by 2.2 times compared to non-edematous children during admission. This may be related to the fact that edematous children lose significant amount of lean body mass and have marginal protein status that precipitated the edema, requiring more time for recovery [24]. Experimental studies on the edema showed that dietary treatment improved edema even before the albumin concentration rose. Among edematous children, there was low plasma zinc concentration and which was strongly associated with nutritional edema and there were significant relationships between plasma zinc concentrations and stunting, skin ulceration, and wasting [11].

However, as earlier weight losses there among severe acute malnourished children after treatment there may be early discharge before cure for some micronutrients like Zink and this may result in recurrence of SAM cases among under five children. When we come to discharge criteria of edematous severe acute malnourished children as loss of edema but not weight gain; however, no cut-off points for weight for the weight after edema.

Another variable that is linked with incidence rat ratio of relapse was age. Being in the age group between 6–11 months increased the incidence of relapse for SAM by 4.74 times compared to those in the age 48–60 months. This may be related to that the target group for screening for the nutritional problem is mainly focusing to the under five-year children and after treatment for severe acute undernutrition there is no follow up at home level if the age is above 60 months as this reason there may relapsed case in this age group but due to age cutoff point for screening will exclude them for admission and diagnosis. This could be related to the fact that age group 6–11 months is the time when complementary feeding is initiated, exposing children to different nutritional and health problems increasing the risk of relapse compared to older ages where children can have different options including family meal.

## Practical implication

In this study as we have seen above Severe acute malnourished children who have edema and early discharge for MUAC results in increasing risk of incidence of relapse. Based on this result we may suggest that existing MUAC for SAM discharge needs rehearsal and it is better to develop discharge additional criteria for SAM children with edema rather than using weight loss as sole criteria.

## Strength

In this study as much as possible we have tried to cover number of health posts with highest number of cases for the last five year with limited number of resources and we have used negative binomial poison regression for this study, to overcome some of the problems of the normal model as for count data poison model has a minimum value of 0 and it will not predict negative values. This makes it ideal for a distribution in which the mean or the most typical value is close to 0.

## Limitation

As this study is retrospective cohort and study design itself bring some limitation and it is better to support this study with prospective cohort to know the sequential order of factors and to identify which factors precedes as cause of relapse. And in this study some important factors were not included due to lack of appropriate or complete registration (e.g., antibiotics, vitamin A, vaccination status, access to health services, standard of living, food security, and access to clean water that may affect relapse to SAM.

## Conclusion

Based on this finding we may conclude that the relapse cases for severe acute undernutrition among under-five children were higher in Ethiopia comparing to the other countries. There were also some factors that increase the incidence of relapse cases; early discharge of MUAC, edema during admission, and age of the child were the linked factors with the Incidence Risk Ratio (IRR) with relapse of SAM cases.

## Recommendation

As there is the highest relapse rate for severe acute malnutrition it is better to have a tracer system for SAM children after discharge from the outpatient treatment program (OTP). And it is better to differ the discharge cut-off point for the weight for age for edematous SAM children and non-edematous SAM children.

In addition to that as the existing discharge point of MUAC is another contributing factor for the incidence of relapse it is an indication for looking a new cutoff point for the discharge of MUAC for SAM under-five children may result in better outcomes after discharge for SAM.

## Supporting information

**S1 Data.**
(ZIP)

## Author Contributions

**Conceptualization:** Abera Lambebo, Deselegn Temiru, Tefera Belachew.

**Data curation:** Abera Lambebo.

**Formal analysis:** Abera Lambebo, Tefera Belachew.

**Investigation:** Abera Lambebo.

**Methodology:** Abera Lambebo, Tefera Belachew.

**Project administration:** Abera Lambebo.

**Resources:** Abera Lambebo.

**Software:** Abera Lambebo, Tefera Belachew.

**Supervision:** Tefera Belachew.

**Visualization:** Abera Lambebo.

**Writing – original draft:** Abera Lambebo.

**Writing – review & editing:** Deselegn Temiru, Tefera Belachew.

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
