## [Decision Letter · Decision Letter 0]

22 Jan 2021

PONE-D-20-40116

Relapse of severe acute malnutrition and associated factors among under five children admitted to health facilities in Hadiya zone, Ethiopia

PLOS ONE

Dear Dr. Lambebo,

Thank you for submitting your manuscript to PLOS ONE. After careful consideration, we feel that it has merit but does not fully meet PLOS ONE’s publication criteria as it currently stands. Therefore, we invite you to submit a revised version of the manuscript that addresses the points raised during the review process.

We look forward to receiving your revised manuscript.

Kind regards,

Claudia Marotta

Academic Editor

PLOS ONE

Journal Requirements:

2. Please provide the names of the 20 selected health posts in Hadiya zone.

3. Please include additional information regarding the survey or questionnaire used in the study and ensure that you have provided sufficient details that others could replicate the analyses. For instance, if you developed the survey or questionnaire as part of this study and it is not under a copyright more restrictive than CC-BY, please include a copy, in both the original language and English, as Supporting Information. If the questionnaire is published, please provide a citation to the (1) questionnaire and/or (2) original publication associated with the questionnaire.

4. Thank you for stating in the text of your manuscript "ethical clearance was be secured by Jimma University Health Research Ethics Review Committee (IHRERC). An official letter was written from Jimma University to the Hadiya Zone health office. Informed written consent was obtained from all health extension workers of selected health posts and woreda health office, Confidentiality of the study documents [were] also ensured according to the declaration of Helsinki ethical code for human subjects." Please also add this information to your ethics statement in the online submission form.

5.In your Data Availability statement, you have not specified where the minimal data set underlying the results described in your manuscript can be found. PLOS defines a study's minimal data set as the underlying data used to reach the conclusions drawn in the manuscript and any additional data required to replicate the reported study findings in their entirety. All PLOS journals require that the minimal data set be made fully available. For more information about our data policy, please see http://journals.plos.org/plosone/s/data-availability.

6.Thank you for stating the following financial disclosure:

 "no"

7. Please amend either the title on the online submission form (via Edit Submission) or the title in the manuscript so that they are identical.

8. Please amend either the abstract on the online submission form (via Edit Submission) or the abstract in the manuscript so that they are identical.

Additional Editor Comments:

dear authors follow reviewer suggestion to improve your paper

Reviewers' comments:

Reviewer's Responses to Questions

**Comments to the Author**

1. Is the manuscript technically sound, and do the data support the conclusions?

Reviewer #1: Yes

Reviewer #2: Partly

Reviewer #3: Yes

2. Has the statistical analysis been performed appropriately and rigorously? 

Reviewer #1: Yes

Reviewer #2: N/A

Reviewer #3: Yes

3. Have the authors made all data underlying the findings in their manuscript fully available?

Reviewer #1: Yes

Reviewer #2: No

Reviewer #3: No

4. Is the manuscript presented in an intelligible fashion and written in standard English?

Reviewer #1: Yes

Reviewer #2: No

Reviewer #3: Yes

5. Review Comments to the Author

Reviewer #1: Authors Wrote an interesting paper on important topic and relevant setting. The article is good but need some improvement also for the reference

Introduction:Africa has a large burden of overall risk factors related to childhood health and development, most of which are of an infective or social origin. Children with malnutrition are "children at risk" to worst health and social outcome (see and citeThe At Risk Child Clinic (ARCC): 3 Years of Health Activities in Support of the Most Vulnerable Children in Beira, Mozambique. Int J Environ Res Public Health. 2018 Jun 27;15(7):1350. doi: 10.3390/ijerph15071350. PMID: 29954117; PMCID: PMC6069480.)

Methods and results: are clear

Discussion: add some risk factors of malnutrition: es HIV, low socio economic level, malaria etc

Reviewer #2: This retrospective study investigated the prevalence of relapse to severe acute malnutrition (SAM) and its associated factors in children under five admitted to health facilities in Hadiya zone (Ethiopia). SAM is an important health problem, particularly in developing countries, and relapse to SAM is one of the important outcomes. Identifying the burden of relapse to SAM in different contexts, its potential risk factors, and consequences are crucial for developing appropriate solutions to this problem. However, the current paper has some serious limitations.

• There have been several publications addressing relapse to SAM in recent years, some of which are from African countries as well as Ethiopia (see references below). Therefore, authors have to provide a convincing rationale for conducting this study.

• The methodology section has many limitations: authors should provide specific definitions for SAM and relapse, briefly illustrate the used treatment protocol, clarify discharge criteria (% of weight gain?), whether there was a system for follow-up/community-based management (CMAM)?, How the 20 health posts in Hadiya zone were selected (not random selection?), authors should briefly describe the collected data.

• Several important factors were not included in the study that may affect relapse to SAM, such as length of time between discharge and relapse, treatment (e.g., antibiotics, vitamin A, ready to use therapeutic food), vaccination status, access to health services, standard of living, food security, and access to clean water.

• The used data from 2004 may not reflect the current sociodemographic situation.

• The discussion is not adequate and does not include data from recent studies, some of which are from Ethiopia (see references below).

• Authors are encouraged to merge the first 3 tables into 1 large table, reporting the frequency for total cases, relapse cases, and no relapse cases for each variable as well as using appropriate statistical tests for comparison.

• The paper is poorly written with poor structure and a lot of language errors.

Stobaugh HC, Mayberry A, McGrath M, Bahwere P, Zagre NM, Manary MJ, Black R, Lelijveld N. Relapse after severe acute malnutrition: A systematic literature review and secondary data analysis. Matern Child Nutr. 2019 Apr;15(2):e12702. doi: 10.1111/mcn.12702.

Abitew DB, Yalew AW, Bezabih AM, Bazzano AN. Predictors of relapse of acute malnutrition following exit from community-based management program in Amhara region, Northwest Ethiopia: An unmatched case-control study. PLoS One. 2020 Apr 22;15(4):e0231524. doi: 10.1371/journal.pone.0231524.

Mengesha MM, Deyessa N, Tegegne BS, Dessie Y. Treatment outcome and factors affecting time to recovery in children with severe acute malnutrition treated at outpatient therapeutic care program. Glob Health Action. 2016 Jul 8;9:30704. doi: 10.3402/gha.v9.30704.

Tadesse E, Worku A, Berhane Y, Ekström EC. An integrated community-based outpatient therapeutic feeding programme for severe acute malnutrition in rural Southern Ethiopia: Recovery, fatality, and nutritional status after discharge. Matern Child Nutr. 2018 Apr;14(2):e12519. doi: 10.1111/mcn.12519.

Kabalo MY, Seifu CN. Treatment outcomes of severe acute malnutrition in children treated within Outpatient Therapeutic Program (OTP) at Wolaita Zone, Southern Ethiopia: retrospective cross-sectional study. J Health Popul Nutr. 2017;36(1):7. Published 2017 Mar 9. doi:10.1186/s41043-017-0083-3

Reviewer #3: A good study, adding to the continuum of studies about relapse of malnutrition in developing countries

However your study should have highlighted the predictors of relapse or at least the recent papers discussing such issue, in the discussion section such as:

Relapse after severe acute malnutrition: A systematic literature review and secondary data analysis

https://www.ncbi.nlm.nih.gov/pmc/articles/PMC6587999/

https://journals.plos.org/plosone/article?id=10.1371/journal.pone.0231524

6. PLOS authors have the option to publish the peer review history of their article (what does this mean?). If published, this will include your full peer review and any attached files.

Reviewer #1: No

Reviewer #2: **Yes: **Elsayed Abdelkreem

Reviewer #3: **Yes: **Antoine AbdelMassih

---

## [Decision Letter · Decision Letter 1]

8 Mar 2021

PONE-D-20-40116R1

Frequency of Relapse of Severe Acute Malnutrition and Associated Factors Among Under Five Children Admitted to Health Facilities in Hadiya Zone, South Ethiopia

PLOS ONE

Dear Dr. Abera

Thank you for submitting your manuscript to PLOS ONE. After careful consideration, we feel that it has merit but does not fully meet PLOS ONE’s publication criteria as it currently stands. Therefore, we invite you to submit a revised version of the manuscript that addresses the points raised during the review process.

We look forward to receiving your revised manuscript.

Kind regards,

Claudia Marotta

Academic Editor

PLOS ONE

Journal Requirements:

Additional Editor Comments (if provided):

Dear Authors, only some minor suggestion to improve your already good paper

Reviewers' comments:

Reviewer's Responses to Questions

**Comments to the Author**

1. If the authors have adequately addressed your comments raised in a previous round of review and you feel that this manuscript is now acceptable for publication, you may indicate that here to bypass the “Comments to the Author” section, enter your conflict of interest statement in the “Confidential to Editor” section, and submit your "Accept" recommendation.

Reviewer #1: All comments have been addressed

Reviewer #2: (No Response)

2. Is the manuscript technically sound, and do the data support the conclusions?

Reviewer #1: Yes

Reviewer #2: Yes

3. Has the statistical analysis been performed appropriately and rigorously? 

Reviewer #1: Yes

Reviewer #2: Yes

4. Have the authors made all data underlying the findings in their manuscript fully available?

Reviewer #1: Yes

Reviewer #2: Yes

5. Is the manuscript presented in an intelligible fashion and written in standard English?

Reviewer #1: Yes

Reviewer #2: No

6. Review Comments to the Author

Reviewer #1: authors wrote an interesting paper and I suggest to accept the paper

Reviewer #2: The authors successfully addressed some reviewers’ comments, but some issues remain.

• In the last paragraph of introduction “Even though; there were some studies on SAM post discharge status there is limitation in addressing relapse case because of methodological and analytical drawbacks”. Authors should provide references to these "some studies" and explain the "methodological and analytical drawbacks" (in introduction or discussion).

• In operational definitions: “Severe acute malnutrition; It is diagnosed by weight for- height below -3 SD of the WHO standards, by a MUAC 11.5 cm and by Clinical sign”. Please, revise to “….MUAC less than 11.5 cm..”, and specify what is the “clinical sign”? bilateral edema?

• In the introduction, authors state that recovery based on MUAC is not used for dischareg “However, they are discharged from the program based on percent of weight gained or weight for height > 70%. During discharge recovery based on MUAC is not used”. However, in operational definitions, they state that MUAC ≥125 mm is one of the discharge criteria “Criteria for discharging children from treatment; weight-for-height/length is ≥–2 Z-scores and they have had no oedema for at least 2 weeks, or mid-upper-arm circumference is ≥125 mm and they have had no oedema for at least 2 weeks”.

• Several important factors were not included in the study and multivariate regression analysis that may affect relapse to SAM, such as length of time between discharge and relapse, treatment (e.g., antibiotics, vitamin A, ready to use therapeutic food), vaccination status, access to health services, standard of living, food security, and access to clean water. The authors should acknowledge this in the study limitations.

7. PLOS authors have the option to publish the peer review history of their article (what does this mean?). If published, this will include your full peer review and any attached files.

Reviewer #1: No

Reviewer #2: **Yes: **Elsayed Abdelkreem

---

## [Author Response · Author response to Decision Letter 1]

9 Mar 2021

all comments from the editors and reviewers were amended and corrected. each parts of comments were attached as separate latter response to reviewers

---

## [Decision Letter · Decision Letter 2]

15 Mar 2021

Frequency of Relapse for Severe Acute Malnutrition and Associated Factors Among Under Five Children Admitted to Health Facilities in Hadiya Zone, South Ethiopia

PONE-D-20-40116R2

Dear Dr. Abera,

We’re pleased to inform you that your manuscript has been judged scientifically suitable for publication and will be formally accepted for publication once it meets all outstanding technical requirements.

Kind regards,

Claudia Marotta

Academic Editor

PLOS ONE

Additional Editor Comments (optional):

dear authors congratulations

Reviewers' comments:

Reviewer's Responses to Questions

**Comments to the Author**

1. If the authors have adequately addressed your comments raised in a previous round of review and you feel that this manuscript is now acceptable for publication, you may indicate that here to bypass the “Comments to the Author” section, enter your conflict of interest statement in the “Confidential to Editor” section, and submit your "Accept" recommendation.

Reviewer #1: All comments have been addressed

Reviewer #2: (No Response)

2. Is the manuscript technically sound, and do the data support the conclusions?

Reviewer #1: Yes

Reviewer #2: Yes

3. Has the statistical analysis been performed appropriately and rigorously? 

Reviewer #1: Yes

Reviewer #2: Yes

4. Have the authors made all data underlying the findings in their manuscript fully available?

Reviewer #1: Yes

Reviewer #2: Yes

5. Is the manuscript presented in an intelligible fashion and written in standard English?

Reviewer #1: Yes

Reviewer #2: Yes

6. Review Comments to the Author

Reviewer #1: no recommendations. the paper can be accept

Authors wrote an interesting paper from interesting setting

Reviewer #2: The authors successfully addressed almost all reviewers' comments. Only a minor comment remains. In operational definitions: “Severe acute malnutrition; It is diagnosed by weight for- height below -3 SD of the WHO standards, by a MUAC 11.5 cm....”. Please, revise to “….MUAC < 11.5 cm..”.

7. PLOS authors have the option to publish the peer review history of their article (what does this mean?). If published, this will include your full peer review and any attached files.

Reviewer #1: No

Reviewer #2: **Yes: **Elsayed Abdelkreem

---

## [Editor Report · Acceptance letter]

17 Mar 2021

PONE-D-20-40116R2 

Frequency of Relapse for Severe Acute Malnutrition and Associated Factors Among Under Five Children Admitted to Health Facilities in Hadiya Zone, South Ethiopia. 

Dear Dr. Lambebo:

I'm pleased to inform you that your manuscript has been deemed suitable for publication in PLOS ONE. Congratulations! Your manuscript is now with our production department. 

Kind regards, 

on behalf of

Dr. Claudia Marotta 

Academic Editor

PLOS ONE